# Alcohol-Tolerant Workplace Environments Are a Risk Factor for Young Adult Alcohol Misuse on and off the Job in Australia and the United States

**DOI:** 10.3390/ijerph20186725

**Published:** 2023-09-07

**Authors:** Sabrina Oesterle, Jennifer A. Bailey, Richard F. Catalano, Marina Epstein, Tracy J. Evans-Whipp, John W. Toumbourou

**Affiliations:** 1Southwest Interdisciplinary Research Center, School of Social Work, Arizona State University, 400 E. Van Buren St., Suite 801, Phoenix, AZ 85004, USA; 2Social Development Research Group, School of Social Work, University of Washington, 9725 Third Ave NE, Suite #401, Seattle, WA 98115, USA; jabailey@uw.edu (J.A.B.);; 3School of Psychology, Deakin University, 1 Gheringhap Street, Geelong, VIC 3220, Australia; 4The University of Melbourne Department of Pediatrics, Royal Children’s Hospital, 50 Flemington Road, Parkville, VIC 3052, Australia

**Keywords:** young adult, alcohol, workplace, cross-national, prevention

## Abstract

The workplace has been understudied as a setting for the prevention of young adult alcohol misuse. This study examined if alcohol-tolerant workplace environments are associated with greater risk for alcohol use and misuse on and off the job among young adults. Data were collected in 2014 from state-representative, sex-balanced samples (51% female) of 25-year-olds in Washington, U.S. (n = 751) and Victoria, Australia (n = 777). Logistic regressions indicated that availability of alcohol at work, absence of a written alcohol policy, and alcohol-tolerant workplace norms and attitudes were independently associated with a 1.5 to 3 times greater odds of on-the-job alcohol use or impairment. Alcohol-tolerant workplace norms were associated also with greater odds of high-risk drinking generally, independent of on-the-job alcohol use or impairment. Associations were mostly similar in Washington and Victoria, although young adults in Victoria perceived their workplaces to be more alcohol-tolerant and were more likely to use alcohol or be impaired at work and to misuse alcohol generally than young adults in Washington. Cross-nationally, workplace interventions that restrict the availability of alcohol, ban alcohol at work, and reduce alcohol-tolerant norms have the potential to prevent and reduce young adults’ alcohol use and misuse on and off the job.

## 1. Introduction

Although alcohol use among adolescents in high-income countries has declined over the last decades [1,2,3], it still is a major contributor to the global burden of disease and mortality [4,5]. The good news is that evidence-based programs and policies now exist to prevent early initiation and use of alcohol in adolescence [6,7], both important risk factors for the development of alcohol use problems later in life. However, there are few efficacious approaches for the prevention of alcohol misuse in young adulthood, despite the fact that alcohol use and misuse peak during this time, making it a critical period for intervention and prevention. The few existing effective approaches for young adults almost exclusively focus on college students [8,9]. More than half of 18- to 24-year-olds do not attend college [10] and may therefore miss out on alcohol and other drug use prevention and treatment services [11]. To address this gap, the present research examined factors in the workplace to explore its potential as an intervention setting for the prevention of alcohol misuse among young adults. We compared data from the U.S. and Australia to probe the generality of the link between the workplace alcohol environment and young adult alcohol misuse across two settings with different alcohol cultures and policy contexts.

Although remote and hybrid work arrangements have become more common in many industries since the COVID-19 pandemic, the workplace is still a promising but mostly overlooked intervention context for young adults that could be relevant for college students and non-college attending young adults alike [12,13,14,15,16]. Most young adults are employed, including college students who tend to work at least part-time. Although remote work arrangements have become more widespread, they are most often part-time and concentrated in certain industries (e.g., computer, business, finance, engineering, sciences, arts, design, and media) [17,18]. The workplace remains an important social context, especially for young adults, as most employees still spend two or three days per week at the office or the work site [17,18] and young workers are overrepresented in food preparation, service, hospitality, and construction [19], i.e., industries that are the least likely to offer remote work options.

The workplace is not only a context that most young adults encounter, but it is also a setting that presents unique risks for alcohol misuse. According to data from the 2002–2003 U.S. National Survey of Workplace Health and Safety (NSWHS) [20], 10.2% of young adult women and 22.0% of young adult men (ages 18–30) were alcohol-impaired at work in the past year due to working under the influence of alcohol or coming to work with a hangover compared to 2.8% of older (ages 31–65) female and 7.4% of older male employees. This is especially true in industries with alcohol-tolerant work cultures, including food preparation and serving; hospitality; arts and entertainment; and construction industries, which are more likely to employ young adults [19,21,22,23,24].

Alcohol-tolerant work environments are concerning not only because they increase the risk for drinking and alcohol-induced impairment at work but also for alcohol misuse in general, including outside of work. A permissive workplace alcohol environment could contribute to the initiation of alcohol use in young adulthood, encourage alcohol misuse among those who did not yet drink heavily, or perpetuate and escalate alcohol problems for those who already had them [25]. To utilize the workplace as a site for alcohol misuse prevention and intervention, we need to identify malleable factors that function as risk mechanisms for alcohol use and misuse at work and possibly also off the job.

Several comprehensive reviews [20,21,26,27] have summarized the state of evidence regarding risk factors for workplace alcohol use. In addition to demographic (e.g., male sex, younger age) and individual characteristics (e.g., rebelliousness, impulsivity), four key aspects of the workplace environment have been shown to increase the risk for alcohol use and alcohol-induced impairment at work: alcohol availability, alcohol policies (and their enforcement by supervisors), work stressors (such as physical hazards and job insecurity), and alcohol norms generated by coworkers’ alcohol use at work (descriptive norms) and their approval of workplace alcohol use (injunctive norms) [20,21,28,29]. It is not yet well understood how these aspects of the workplace environment work together in creating a risk context for alcohol use and whether each is uniquely associated with alcohol misuse, particularly among young adults.

We also know little about the generality of the link between workplace alcohol environments and alcohol use across policy contexts or cultures, limiting a more universal understanding of underlying processes that could improve intervention development and implementation [21]. Although alcohol use is highly permissible in most Western countries and not considered a drug, countries differ in their alcohol policy orientations and cultures, especially with respect to youth alcohol use. U.S. policies, for example, tend to be restrictive, aiming for abstinence, with an older legal drinking age and more restrictive laws toward alcohol sales. Other countries, such as Australia, have more permissive policies focusing on harm reduction. In line with these policy orientations, rates of alcohol misuse and use disorder among youth and young adults are higher in Australia compared to the U.S. [30,31].

The present study aimed to fill several of the gaps in the existing understanding of the link between the workplace alcohol environment and young adult workers’ alcohol misuse off and on the job by comparing survey data from state-representative samples of 25-year-olds in Washington, U.S. and Victoria, Australia. We examined the link between three key aspects of the workplace alcohol environment (alcohol availability, policy, and norms) and young adults’ alcohol use or impairment at work as well as their high-risk drinking in general.

We expected that young adults in Victoria would report more tolerant alcohol environments at work than young adults in Washington and, consequently, more prevalent workplace alcohol use and impairment. We hypothesized that all three environmental workplace factors would be associated with alcohol use and impairment at work but only indirectly with alcohol misuse in general. We expected these associations to be similar in Victoria and Washington based on evidence from prior analyses of this dataset [30] and other cross-national studies [32]

## 2. Materials and Methods

### 2.1. Data and Sample

Data came from the International Youth Development Study (IYDS), a cross-nationally matched longitudinal cohort study of state-representative samples of youth from Washington State in the U.S. and Victoria in Australia. Washington and Victoria were chosen due to their distinct alcohol and drug policy orientations and their similarities in population size, urbanization, educational participation, prosperity, and ethnic diversity. Youth in the present study were initially recruited and surveyed in 2002, with parental consent, when they were in seventh grade (average age 13 years) and re-surveyed at ages 14, 15, and 25 (in 2014).

The study used identical sampling and survey administration strategies in both states as described previously [33]. A two-stage cluster sampling approach was used to randomly select, first, public and private schools in each state containing grades 5, 7, or 9 and, second, a target classroom within each school. The 305 participating schools (153 in Washington and 152 in Victoria) were representative of all schools in each state with respect to school type, economic disadvantage, and student diversity except that private schools were underrepresented in the Washington school sample and schools with medium proportions (28–43%) of students from low-income families were overrepresented in Victoria. The present study used data only from the original seventh grade cohort.

Of eligible seventh-grade students, 78.4% (n = 961; mean age = 13.1 years, *SD* = 0.4) in Washington and 75.6% (n = 984; mean age = 12.9 years, *SD* = 0.4) in Victoria were recruited into the study with 51% in both states being female.

In total, 65% of Washington students were White, 16% Hispanic/Latino, 6% Asian/Pacific Islander, 6% Native American, 4% African American, and 3% other. In Victoria, race and ethnicity were conceptualized differently, with the majority of youth identifying as Australian (91%) regardless of their racial or cultural ancestry; 6% identified as Asian/Pacific Islander, 1% as Aboriginal, 0.7% as African, 0.4% as Spanish/Hispanic/Latino, and 1% as other. At age 25 (2014), 87.5% of the original participants in Washington (n = 841) and 88.0% of the original participants in Victoria (n = 866) completed the survey. For the present study, the analyses were restricted to those who reported at age 25 that they were employed (currently or in the past year; 89.3% [n = 751] in Washington and 89.7% [n = 777] in Victoria). 

### 2.2. Survey Instrument and Measures

All data were self-reports collected using an adapted and extended version of the Communities that Care Youth Survey, which has shown good reliability and validity in the U.S. and Australia [30,34]. Questionnaires were administered in classrooms during a 50- to 60-minute period during the school years. Students in Washington received $10 upon survey completion, whereas students in Victoria received a small pocket calculator upon return of consent forms. The age 25 (2014) survey was administered online. Upon completion, respondents received an incentive of about $40 in local currency. All procedures were approved by the University of Washington Human Subjects Institutional Review Board and by the Royal Children’s Hospital Ethics in Human Research Committee and the University of Melbourne Human Ethics in Research Committee.

Except for covariates, all measures used in this study were based on data collected at age 25.

*Workplace alcohol environment.* Participants were asked to think about the workplace at which they spent the most time during the past 12 months and to indicate whether alcohol was available for employees (a) through staff club or cafeteria, (b) at special work functions and occasions, (c) during regular end of the day drinks, (d) for use by management, and (e) in any other way. For analyses, availability in any situation was combined into one category (1 = alcohol available at work, 0 = not available). Participants were also asked whether their workplace had a written policy regarding alcohol use and being under the influence of alcohol at work and if the policy completely banned alcohol (yes, no, do not know).

To measure behavioral norms, respondents were asked to report (choosing from 1 = “never” to 8 = “40 or more times”) how often coworkers they typically interacted or worked with each day used alcohol during the workday, including lunch or other breaks, or came to work drunk or under the influence of alcohol during the past 12 months. For analyses, responses were grouped into “never” compared to “1–2 times or more often”. Participants also were asked to describe their workplace’s attitude toward drinking or being under the influence of alcohol at work. Because few participants reported that their workplace encouraged or tolerated alcohol use even if frequent, they were combined for analyses with those who said that alcohol was tolerated if not frequent.

*Workplace alcohol use or impairment*. Respondents reported (choosing from 1 = “never” to 8 = “40 or more times”) how often in the past 12 months they had (a) drunk alcohol at work (including lunch and breaks), (b) drunk four or more alcoholic drinks within an hour of starting work, (c) worked while under the influence of alcohol, (d) come to work with a hangover, and (e) drunk alcohol after work at their place of work before coming home. Analyses assessed if respondents had engaged in *any* workplace alcohol use or had experienced any impairment at least once in the past year.

*General high-risk drinking*. High-risk drinking in general was measured using the 10-item Alcohol Use Disorders Identification Test [35]. Analyses compared those who met criteria for hazardous or harmful drinking in the past year (total score of 8 or higher) to those who did not [36,37].

*Covariates*. Analyses were adjusted for sociodemographic and economic characteristics of the respondents (sex, race, four-year college degree, full-time student, parent, married, and current financial problems) as well as employment characteristics (i.e., months of full-time employment in the past year). Those employed in sales and retail; food and hospitality; construction and extraction; arts, media, sports, and entertainment; or transportation and storage industries were coded as working in a high-risk industry.

Because underage drinking is a risk factor for later alcohol problems and high-risk drinking in young adulthood and could explain observed relationships between workplace alcohol environment and on- and off-the-job drinking at age 25, analyses also adjusted for prospectively measured adolescent alcohol use (average past-year frequency of drinking self-reported in grades 7, 8, or 9).

### 2.3. Analysis

We estimated two multivariable logistic regression models, first using each workplace alcohol environment variable, one at a time, as an independent variable in the model (Model 1), and then accounting for all of them simultaneously (Model 2) to examine their unique and additive associations with workplace and general alcohol use. Both models also adjusted for all covariates. In addition, concurrent general high-risk drinking was included when workplace alcohol use or impairment was the dependent variable to test the extent to which young adults were more likely to use alcohol at work, not because their workplace alcohol environment was permissive, but because they engaged in high-risk drinking in general. Analyses of general high-risk drinking included workplace alcohol use or impairment to test the extent to which any observed association between workplace alcohol environment and general high-risk drinking was likely indirect, mediated through the increased likelihood of alcohol use at work in an alcohol-tolerant work environment. Finally, to test the extent to which associations differed between Washington and Victoria, we estimated an interaction term between study site (0 = Washington, 1 = Victoria) and each workplace alcohol environment variable, one at a time, in the pooled sample. All analyses were conducted using IBM SPSS Statistics version 26 [38].

Because attrition through age 25 was low (12%), almost identical in both states, and item-nonresponse at age 25 was minimal (i.e., one or two cases for a given variable), the risk of bias due to missing data was low. A comparison of the analysis samples with the original samples in both states (see Appendix A) indicated that they did not differ on a range of baseline characteristics including sex, racial/ethnic composition, age, parental education, religious service attendance, grades in school, and alcohol use. All analyses were, thus, conducted using the complete-case dataset. Due to varying rates of missing data in the multivariable analyses, sample sizes ranged from 764 to 765 participants in Victoria and 723 to 727 in Washington

## 3. Results

The samples of employed young adults in Washington and Victoria were similar with respect to age and sex but differed with respect to a few other characteristics (Table 1). Young adults in Washington were less likely than those in Victoria to be in a majority racial/ethnic group (White in Washington and Australian in Victoria) and to have a four-year college degree, but significantly more likely to be married, to be a parent, and have financial difficulties. The samples were similar with respect to the average number of months young adults worked full-time and the proportion who attended school full-time and worked in an industry with high-risk for workplace alcohol use.

As hypothesized, young adults in Victoria were almost twice as likely as those in Washington to engage in high-risk drinking (Table 2). They were also about one and half times as likely to have used alcohol in adolescence and had used it more frequently. Young adults in Victoria were almost twice as likely than those in Washington to drink at work during work hours, almost three times as likely to drink at work after work hours, and 25% more likely to have come to work with a hangover. Similar proportions of employed young adults in both states worked while under the influence of alcohol.

As hypothesized, workplace alcohol environments of young adults in Victoria were significantly more tolerant compared to those in Washington (Table 3). Young adults in Victoria were about twice as likely to report that alcohol was available at their workplace. In both states, alcohol at work was most common at special work functions and occasions, but with significantly higher prevalence in Victoria than Washington. Young adults in Victoria also were less likely than those in Washington to say that their workplace had a written alcohol policy, including one that completely banned alcohol, but were more likely to say that they did not know if there was a policy. Young adults in Victoria also were more likely than those in Washington to report that their coworkers used alcohol at work or came to work alcohol-impaired and they perceived attitudes toward alcohol use in their workplaces to be more tolerant than in Washington. All dimensions of the workplace alcohol environment were strongly associated with each other (see Appendix A).

Regression analyses indicated that each dimension of the workplace alcohol environment on its own was associated with increased odds of alcohol use and impairment at work, even after controlling for concurrent high-risk drinking in general (Figure 1, Model 1), and most had also an independent influence when all workplace factors were considered together (Model 2). For the most part, relationships were similar in strength in Victoria and Washington and were only somewhat attenuated in Model 2 (see Appendix A).

The availability of alcohol at work was a strong risk factor for alcohol use or impairment at work. The statistical interaction test suggested that this association was about two and a half times stronger in Victoria than Washington (AOR = 2.49; 95% CI: 1.44; 4.40; *p* = 0.002). Young adults who said alcohol was available at work were five times as likely to use alcohol or be impaired at work as those who said it was not available if they lived in Victoria (95% CI for AOR: 3.40; 7.51, *p* = 0.000) and two times as likely if they lived in Washington (95% CI for AOR: 1.26; 3.22, *p* = 0.003).

A workplace that did not have a written alcohol policy, especially one that did not completely ban alcohol, increased the odds of young adults’ alcohol use or impairment at work three-fold (Figure 1, Model 1). The strength of this association was slightly attenuated in Model 2. Although the state-specific analyses suggested somewhat stronger relationships in Victoria than Washington, interaction tests (see Appendix A) did not provide strong evidence against the null hypothesis of equal associations.

Workplace alcohol norms increased the odds of workplace alcohol use and impairment in both states and there was little evidence according to the statistical interaction tests (see Appendix A) that the data were not consistent with the null hypothesis of equal associations in both states. Young adults who thought their workplace tolerated or even encouraged alcohol use were almost five times as likely to use alcohol at work as those who thought that alcohol was not acceptable in their workplace (Figure 1, Model 1). Even perceiving the workplace to be only discouraging of alcohol use increased the odds of drinking and impairment at work almost three-fold. In Model 2, the impact of attitudinal norms on alcohol use or impairment at work was about halved. Young adults who said their coworkers used alcohol at work were about four times as likely to drink at work as those who reported that their coworkers did not use alcohol (Figure 1, Model 1); this association was slightly attenuated in Model 2.

As expected, the workplace alcohol environment was less consistently associated with general high-risk drinking after adjusting for concurrent workplace alcohol use or impairment. There was little evidence that the availability of alcohol at work, the absence of at least some workplace alcohol policy, or the absence of a policy that completely banned alcohol were associated with general high-risk drinking (Figure 2). In Washington, however, not knowing if the workplace had a written alcohol policy or whether an existing policy banned alcohol doubled the odds of general high-risk drinking (AOR = 2.00 [1.15; 3.49], *p* = 0.015), even after adjusting for workplace alcohol use or impairment and the other workplace dimensions. The interaction test supported the interpretation that this association was significantly weaker in Victoria (AOR = 0.41 [0.21; 0.80], *p* = 0.009). Behavioral and attitudinal workplace alcohol norms, on the other hand, were independent risk factors for general high-risk drinking, even beyond alcohol use or impairment at work (Figure 2). They increased the odds of high-risk drinking between 42% and 75%. The interaction tests did not provide strong evidence against the null hypothesis of equal associations in the two states (see Appendix A).

## 4. Discussion

This cross-national study set out to fill several of the gaps in the understanding of the link between a tolerant workplace alcohol environment and young adults’ alcohol use off and on the job. Results suggested that all three dimensions of the workplace alcohol environment considered in this study—availability of alcohol in the workplace, absence of a written alcohol policy, and alcohol-tolerant norms and attitudes—were independently associated with greater the risk for on-the-job alcohol use or impairment. Only alcohol-tolerant workplace norms, but not alcohol availability or policy, were associated with greater young adults’ risk for hazardous drinking in general, including outside of work, after adjusting for their on-the job alcohol use. This finding may reflect that young adults likely also socialize outside of work with alcohol misusing coworkers.

As expected, associations were for the most part similar in Washington and Victoria, with one primary exception. Availability of alcohol at work was 2.5 times as strong of a risk factor for workplace alcohol use or impairment in Victoria as it was in Washington, beyond the risk posed by the absence of an alcohol policy and alcohol-tolerant norms. Young adults in Victoria were particularly more likely than those in Washington to report that alcohol was available at special work functions and occasions (see Table 3), possibly indicating that drinking at work is part of job expectations more so in Australia than in the U.S.

Prior research is not yet sufficient to develop effective workplace alcohol interventions [39]. The findings from this study add to prior research e.g., [20,21,28,29] in four ways. First, by examining three key aspects of the workplace alcohol environment together, this study increased understanding of the total and unique associations of various workplace features that will help inform the development of workplace alcohol interventions. Second, by considering high-risk drinking in general as well as on-the-job drinking and impairment, this study examined the extent to which the workplace alcohol environment could have broader implications for alcohol misuse prevention. A doubling of the odds of on-the-job drinking or impairment due to a tolerant workplace alcohol environment, as was found in this study, could be associated with a significantly higher risk for general high-risk drinking indirectly because of their strong association. The current study fills in knowledge gaps for young adults [12] who reported using alcohol or being alcohol-impaired at work were almost nine times as likely to also report high-risk drinking in general compared to those who did not use alcohol at work. Third, by using prospectively measured adolescent risk for alcohol misuse and by accounting for demographic, individual, and employment differences, the analyses controlled for potential selection and confounding associations in a way often not possible in prior research (e.g., [29]). The study, thus, strengthened the case for causal interpretations. Fourth, by comparing the link between workplace alcohol environment and workplace drinking in the U.S. and Australia, findings from this study have generalizable implications for contexts with different policy settings and alcohol cultures. For the most part, a permissive alcohol environment at work was associated with greater risk of on-the-job drinking or impairment to a similar degree in Victoria and Washington

The study had several limitations. First, data for this study were collected in 2014 and may not completely reflect contemporary workplace alcohol environments. Workplaces all around the world have seen an increase in remote and hybrid work arrangements since the COVID-19 pandemic began in 2020. To what extent this change impacted workplace alcohol cultures and policies is not yet known and an important topic for future investigation. However, young adults remain overrepresented in occupations that are less likely to offer remote work options and that are in industries with alcohol-tolerant work cultures, such as food preparation and serving; hospitality; arts and entertainment; and construction industries. The results of the present study, therefore, remain relevant for today’s young workers. Furthermore, the focus of the study were the associations between risk factors in workplace alcohol environments and young adult alcohol use, as well as cross-national variation in these associations. Although prevalence of both alcohol use and levels of risk may have changed, associations between the two most likely did not. The present study did not examine coming to work with a hangover separately from the other alcohol-related workplace behaviors. Because this was the most common type of workplace impairment in both states, it would be a worthwhile objective for future research to examine to what extent the associations between alcohol-related workplace factors may differ for different types of alcohol-related workplace behaviors. Second, data were self-reports and alcohol use may have been underreported due to social desirability. However, the study was primarily concerned with estimating associations between variables rather than prevalence, which are less likely to be affected by bias due to self-reports. Third, measures of primary variables of interest were contemporaneous. Associations between the workplace alcohol environment and alcohol use can, therefore, only be interpreted as correlational. However, analyses adjusted for concurrent alcohol use or impairment at work when examining general high-risk drinking and vice versa, thus, controlling for their strong mutual correlation to prevent falsely attributing observed associations to the workplace alcohol environment. Furthermore, controlling for a range of possible confounders, including prospectively measured alcohol use in adolescence, further increased confidence in the conclusions. Finally, the comparison of data from two states in two countries did not allow for a direct evaluation of the impact of different state or national-level alcohol policies. The two national contexts were used primarily to develop broad hypotheses about possible state differences and findings need to be considered in the context of other studies.

## 5. Conclusions

Given a general lack of existing evidence-based programs for young adults who are not in college, this study’s findings have important implications. Results suggest that the workplace promises to be an intervention setting with potentially high impact for the prevention and reduction in young adult alcohol misuse even in the new post-COVID-19 pandemic context. Workplace interventions that restrict alcohol availability, or better still, completely ban alcohol at work, and also aim to reduce alcohol-tolerant norms among employees have the potential to significantly contribute to the prevention and reduction in young adult workers’ alcohol use on and even off the job.

## Figures and Tables

**Figure 1 ijerph-20-06725-f001:**
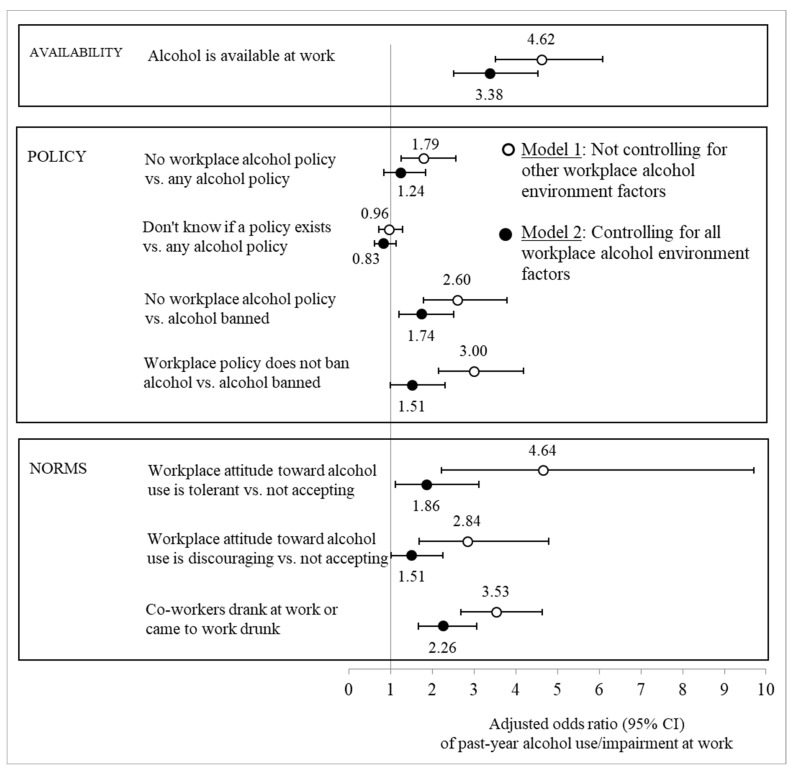
Adjusted odds ratios (with 95% confidence intervals) of alcohol use or impairment at work in the past year given workplace alcohol environment factors. Note: Based on logistic regression estimates adjusted for concurrent general high-risk drinking in the past year, covariates (sex, race, four-year college degree, full-time student, parent, married, current financial problems, months of full-time employment in the past year, high-risk industry, and adolescent alcohol use).

**Figure 2 ijerph-20-06725-f002:**
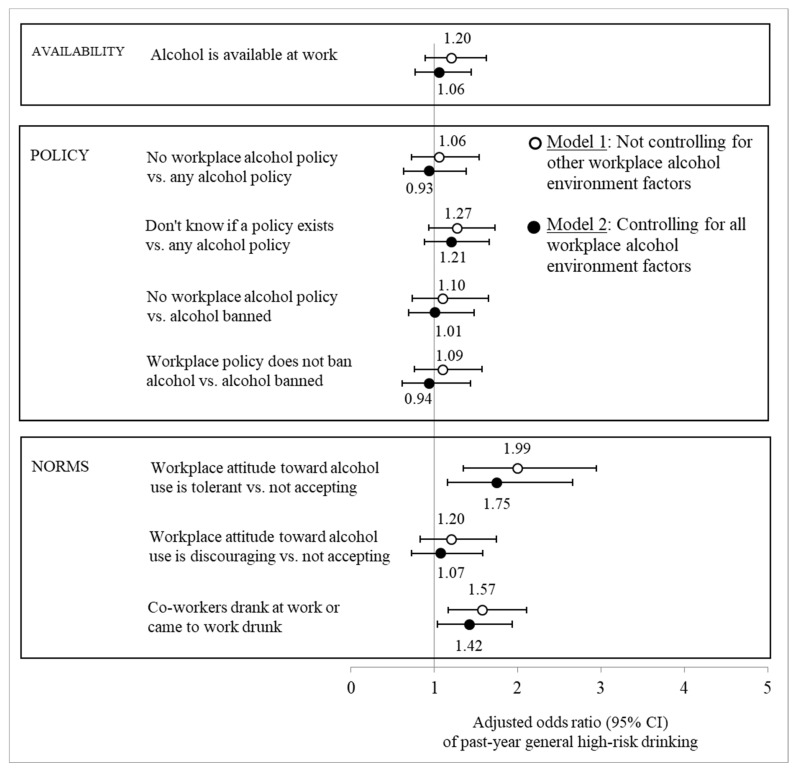
Adjusted odds ratios (with 95% confidence intervals) of general high-risk drinking in the past year given workplace alcohol environment factors. Note: Based on logistic regression estimates adjusted for concurrent alcohol use or impairment at work in the past year, covariates (sex, race, four-year college degree, full-time student, parent, married, current financial problems, months of full-time employment in the past year, high-risk industry, and adolescent alcohol use).

**Table 1 ijerph-20-06725-t001:** Characteristics of employed young adults in Victoria and Washington.

	Victoria	Washington	Difference	
	n = 777 ^a^	n = 751 ^a^	Chi-Square	*p*
Age in years				
24, %	52.4	32.5	64.7	0.000
25, %	44.0	61.9		
26, %	3.6	5.6		
Male, %	48.3	46.7	0.4	0.551
White/Australian, %	91.4	76.7	61.7	0.000
Four-year college degree, %	44.3	29.6	35.2	0.000
Full-time student, %	12.2	10.8	0.8	0.378
Married, %	6.2	22.2	80.7	0.000
Parent, %	10.7	25.6	57.2	0.000
Financial difficulties, ^b^ %	23.1	35.8	29.7	0.000
Months of full-time employment in the past year (35+ hours/week), mean (SD)	7.8 (5.1)	8.1 (4.6)	1.3 ^d^	0.180
High-risk industry ^c^	42.0	41.0	0.2	0.679

Notes: ^a^ Due to missing data, sample sizes ranged from 773 to 777 in Victoria and 744 to 751 in Washington. ^b^ Just getting by or finding financial situation quite or very difficult. ^c^ Employed in industries with high risk for workplace alcohol use includes sales and retail; food and hospitality; construction and extraction; arts, media, sports, and entertainment; and transportation and storage industries. ^d^ *t*-test for difference in means.

**Table 2 ijerph-20-06725-t002:** General high-risk drinking and workplace alcohol use among employed young adults in Victoria and Washington.

	Victoria	Washington	Difference	
	n = 777 ^a^	n = 751 ^a^	Chi-Square	*p*
Age 25 general alcohol misuse				
Hazardous or harmful drinking, ^b^ %	30.5	17.5	35.3	0.000
Age 25 alcohol use or impairment at work				
Drank at work during work hours, %	26.3	14.2	34.4	0.000
Drank at work after work hours, %	23.2	8.3	63.2	0.000
Worked while under the influence of alcohol, %	14.7	11.6	3.1	0.079
Came to work with a hangover, %	46.5	37.2	13.5	0.000
Any workplace alcohol use or impairment, %	57.1	42.5	32.6	0.000
Adolescent alcohol use				
Had alcohol in grades 7, 8, or 9, %	71.5	45.4	107.3	0.000
Average past-year frequency of alcohol use grades 7–9, ^c^ mean (SD)	2.8 (1.6)	1.8 (1.1)	13.2 ^d^	0.000

Notes: ^a^ Due to missing data, sample sizes ranged from 773 to 777 in Victoria and 744 to 751 in Washington. ^b^ Total score of 8 or higher on Alcohol Use Disorders Identification Test (AUDIT). ^c^ 1 = never, 2 = 1–2 times, 3 = 3–5 times, 4 = 6–9 times, 5 = 10–19 times, 6 = 20–29 times, 7 = 30–39 times, 8 = 40 or more times. ^d^ *t*-test for difference in means.

**Table 3 ijerph-20-06725-t003:** Workplace alcohol environment of employed young adults in Victoria and Washington.

	Victoria	Washington	Difference	
	n = 777 ^a^	n = 750 ^a^	Chi-Square	*p*
Alcohol is available at work, %	46.6	21.8	106.04	0.000
Through staff club or cafeteria, %	4.5	1.6	10.79	0.001
At special work functions and occasions, %	33.5	12.7	92.49	0.000
During regular end of the day drinks, %	12.6	6.4	17.03	0.000
For use by management, %	1.8	1.1	1.45	0.228
Another way, %	4.0	4.1	0.02	0.887
Workplace has a written alcohol policy, %	55.7	69.4	31.06	0.000
Alcohol is completely banned, %	30.8	54.8	91.06	0.000
Alcohol is not completely banned, %	24.9	14.6		
No policy, %	15.1	11.6		
Do not know, %	29.2	19.0		
Workplace attitude toward drinking or being under the influence of alcohol at work				
Not acceptable, %	71.7	81.3	23.68	0.000
Discouraged, %	14.5	11.5		
Tolerated if not frequent, %	10.7	5.9		
Tolerated even if frequent, %	1.8	0.9		
Encouraged, %	1.3	0.4		
Coworkers drank at work or came to work drunk or under the influence of alcohol, %	31.9	26.0	6.41	0.011

Note: ^a^ Due to missing data, sample sizes ranged from 775 to 777 in Victoria and 747 to 750 in Washington.

## Data Availability

The data presented in this study are available on request from the second author, Dr. Bailey.

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
