# Peer review of "Alcohol-Tolerant Workplace Environments Are a Risk Factor for Young Adult Alcohol Misuse on and off the Job in Australia and the United States"

_ijerph, 2023, doi:10.3390/ijerph20186725_

Round 1

Reviewer 1 Report

Alcohol misuse is prevalent among adolescents and young adults and can lead to serious health consequences later in life. While evidence-based prevention programs for adolescents are often implemented in schools, and similar interventions for young adults are provided in colleges and universities, a large proportion of young adults in their mid-20s do not attend college, instead going directly from high school to work. Thus, many young adults are excluded from potentially helpful interventions, making workplaces a tempting target for instituting efforts to prevent alcohol misuse.

Alcohol misuse on the job (e.g., being under the influence while working or coming to work experiencing aftereffects of drinking) is also prevalent in some industries (e.g., construction, food and drink service, entertainment) that tend to employ young people. Workplaces differ in the extent to which alcohol use is tolerated, and it is conceivable that more alcohol tolerant workplace policies and attitudes might be associated with greater risks for alcohol misuse both on and off the job. Exploring associations between workplace tolerance toward alcohol use and both misuse on the job and risky drinking in general therefore is worthwhile. This paper addresses several gaps in our understanding by investigating associations between three aspects of alcohol tolerance in the workplace (i.e., availability of alcohol at work, existence and enforcement of policies on alcohol use at work, and norms for alcohol use) and both misuse on the job and risky consumption off the job.

The data source is a two-country longitudinal study begun when participants were age 13 (in 7th grade), with one data collection point at age 25. Participants were recruited from Washington state in the US and Victoria state in Australia, with the aim of comparing cultures with differing norms and policies related to alcohol use. Data for the current study were primarily collected at age 25, although one strength is the availability of data on frequency of drinking during adolescence, which served as a covariate. The sample size is large (751 from Washington; 777 from Victoria), diverse and comparable across countries on key variables (e.g., employment in a high-risk industry, number of months of full-time employment, sex). The Washington sample skewed older, but the difference is unimportant given the limited age range (24-26 y).

Appropriate measures were used to assess key variables and the analytic methods were appropriate. I'm impressed by the thought given to controlling for relevant variables (e.g., concurrent high-risk drinking when examining workplace use/impairment and workplace use/impairment when examining high-risk drinking) and to including interaction terms to explore possible state by exposure influences.

The paper is very clearly written, the tables and especially the figures are easy to interpret, and the numerous supplemental tables provide a complete picture of the data. Kudos. The discussion does an excellent job of tying the findings to existing research, of accounting for limitations, and of examining implications for workplace prevention programs.

I have a few suggestions for minor revisions:

1. I agree that 12% overall attrition over 12 years is potentially ignorable. However, the effective attrition (i.e., number of participants in the analytic sample vs. the original number of participants in 7th grade) is 22% in Washington and 21% in Victoria. Assuming the state samples were comparable at age 13, I would like to see a supplemental descriptive table comparing baseline characteristics of the 417 participants not retained for analysis (210 from Washington, 217 from Victoria) in this study with the 1528 retained. That would provide readers a way to assess whether selection bias might be a factor. As the table would be purely descriptive, I suggest following STROBE guidelines (https://doi.org/10.1371/journal.pmed.0040297) and not testing for differences.

2. Abstract, line 24: "Associations were [for] mostly similar ..." I suspect the sentence originally read 'were, for the most part, similar' and was then shortened, making 'for' unnecessary.

3. Methods, line 197: You describe your analyses as multivariate, yet as I understand the process, each analysis considered only one outcome. In that case, the analyses were 'multivariable' (i.e., one outcome, more than one exposure or covariate). Please correct.

4. Methods, line 218 and Results, line 223: You use 'gender' in both sentences, yet talk only about males, which suggests a binary coding (male/female). Did you query gender (i.e., allow for more options than biological sex) or did you query sex by limiting responses to male or female? If the latter, please substitute 'sex' for 'gender' where appropriate. I suspect line 186 is correct - i.e., that the variable measured sex.

5. Methods, lines 201/202 and 204/205: I realize that you're using the word 'predicting' (which implies temporality) strictly in the statistical sense, as exposures and outcomes were measured simultaneously. To avoid confusion, I suggest substituting 'examining' for 'predicting' in both instances.

Author Response

  1. As requested by the reviewer, we added a descriptive supplemental table (new supplemental Table 1) showing baseline characteristics for the analysis sample (those surveyed at age 25 and who were employed) compared to the original sample. Results indicate that the analysis sample did not differ dramatically from the original sample with respect to sex, racial/ethnic composition, age, parental education, religious service attendance, grades in school, and alcohol use. We added a description of these results and reference to the supplemental table to the Analysis section.

  2. We have deleted the word "for" in the sentence in the abstract, line 24, as it was not needed.

  3. In the Methods section, we have replaced the word "multivariate" with "multivariable" as recommended to indicate that the regression analysis included multiple independent, not multiple dependent variables. 

  4. The reviewer is correct that we measured binary biological sex (male/female) not gender. We have replaced all reference to “gender” with the word “sex” in the manuscript. 

  5. We have replaced the word "predicting" throughout the manuscript with appropriate phrasing that does not imply temporality or causality, such as "associated with" and "examining." We replaced the word "predictor" with "independent variable."  

Reviewer 2 Report

The study addresses a gap in the literature regarding workplace factors related to alcohol use at work and heavy or problematic drinking. The study includes observational data from Australia and the US. The results indicate that policies, norms, and alcohol availability are associated with employee alcohol-related impairment or use.

The authors address an area that is important, scientifically interesting, and little studied. I don't work on workplace factors. I do work on substance use in this developmental period.

This is a well-written article. I have several questions, a suggestion, and a concern that can be easily addressed.

1 Causal Inferences

The main concern is the causal inferences made in the discussion and conclusion. The authors recognize the limits of their study design for supporting causal claims in a passage on limitations (page 11 line 358). The first paragraph of the discussion attributes increases in drinking to the investigated independent variables. The study design alone does not support the implication that the IVs are responsible for the totality of the elevated drinking associated with the IVs. An increased rationale that is triangulated with the results could potentially justify the causal inferences.

The conclusion regarding interventions is only justified if the workplace factors are causal and substantive in their magnitude of effect. This is particularly the case for the statement on line 393. Conclusions about relative strength are not supported because of the comments made above, and the comparators are not described. If the identified factors are the strongest, what are they stronger than? That was not clear.

As a reader, I wondered to what extent people who will tend to drink more are going to select or be filtered into jobs where that is more permissible, creating a confound.

2 Hangover Vs Use on the Job

The main outcome variable is workplace alcohol use or impairment. Drinking on the job is combined with having a hangover while on the job. Hangover was the most common type of alcohol-related impairment. Particularly given the differential susceptibility for hangovers that might be related to drinking patterns and risks for AUD/drinking-related risks, and there seems like there would be very different processes involved in hangover and drinking on the job, why isn't hangover and drinking at work analyzed separately?  I assumed that the policies and norms items addressed drinking on the job rather than having a hangover at work.

3. Description.

There is a reference to the full description of the sample ascertainment methods. It would be useful to provide a sketch of the sample ascertainment methods.

4. Table 2 notes

c and d footnotes appear to be missing.

Author Response

  1. The reviewer was concerned about the causal inferences made in the Discussion and Conclusions given the mostly cross-sectional nature of the data. In recognition of this concern, we have changed the language in the manuscript to indicate that our analyses were correlational, e.g., replacing “increased” with “associated with” and “effects” with “associations.”

  2. The reviewer pointed out that a statement (previously on line 393) in the Conclusions was confusing since we did not describe the comparators for our statement about factors that could have the strongest influence on young adult alcohol use. We agree that this was unclear and have changed this sentence to: “Workplace interventions that restrict alcohol availability, or better still, completely ban alcohol at work, and also aim to reduce alcohol-tolerant norms among employees have the potential to significantly contribute to the prevention and reduction of young adult workers’ alcohol use on and even off the job.”

  3. The reviewer wondered “to what extent people who will tend to drink more are going to select or be filtered into jobs where that is more permissible, creating a confound.” We agree that this is a possibility. Our analyses addressed this concern as they adjusted for both drinking in adolescence and high-risk industry, thus, taking their association into account when examining alcohol use both on and off the job.

  4. The reviewer wondered why we did not analyze having a hangover while on the job separately from drinking at work given that coming to work with a hangover was the most common type of alcohol-related impairment and could potentially have differential associations with the alcohol-related workplace factors examined in this study. Although this is a possibility, we did not have hypotheses about different associations between workplace environment factors and drinking at work during work hours versus after hours or working while under the influence of alcohol versus coming to work with a hangover. Our main objective in this study was to contrast how different aspects of the workplace alcohol environment, including alcohol availability, policy, and norms, were associated with alcohol use or impairment at work broadly defined and compared to harmful drinking more generally and in the two states. To facilitate the complexity of the analyses and avoid multiple statistical tests (and keep Type I error in check), we combined all workplace alcohol use and impairment measures to create a single outcome measure. However, analyses that separate out different aspects of workplace alcohol use and impairment would be a worthwhile objective for future analyses and we have added a statement to the discussion section suggesting this direction for future work.

  5. The reviewer requested more detail on the sample ascertainment methods, which we added to the text, describing the 2-stage sampling procedure, which randomly selected first schools and then a classroom within each school from which students were recruited.

  6. The reviewer noted that footnotes c and d to Table 2 might be missing. We checked and found that they were included in the submitted manuscript but were shown on the following page in the system-generated pdf version of the manuscript, which may have caused readers to miss this information. No changes were, therefore, necessary in the revised version of the manuscript.

Reviewer 3 Report

Thank you for the opportunity to review this manuscript. Let me start with the positive aspects of the manuscript. Comparative research between two countries is always very interesting and I was curious about the results presented by the authors. I believe that these studies are really important and can be used in the planning of preventive actions and even in the programming of an appropriate anti-alcohol policy.

Unfortunately, I also have some doubts.

The selection of respondents for the research is unclear to me. I believe that the authors should present the sampling procedure and not just refer to other studies. Were the same respondents in all measurements? How were they reached, both in traditional and online research? How many people did not respond to retesting? And similar data. This has been partly described, however, the reader should not look in the references for the research methodology he reads about in the article. I am writing about it because the sample is impressive and I am very interested in how the authors managed to get so many respondents. I know it's not easy.

I also wonder why write about these studies, which were started in 2002, since their results are not reported here, and therefore seem irrelevant. It would probably be better to focus on the final selection of the sample and a solid description of this procedure.

The manuscript presents the results of research carried out in 2014. Taking into account the dynamics of the modern world, there have been significant changes over the last decade in both attitudes towards alcohol and reasons for reaching for such beverages.

The discussion section is incomplete. Discussion, i.e. comparing the conducted research with other similar research: indicating common elements, differences, possibilities, etc. I believe that in such research it is an important element of scientific work.

Research is really interesting, but the form of their presentation should be at a much higher level.

When it comes to the technical side, there is also a lot to improve: incorrect footnotes, References section, terrible punctuation or even no punctuation).

Round 2

Reviewer 3 Report

The corrections made have improved the quality of the manuscript. It can be published in this form. Thank you for following most of my comments.